# Improving the Performance of the ToGoFET Probe: Advances in Design, Fabrication, and Signal Processing

**DOI:** 10.3390/mi12111303

**Published:** 2021-10-23

**Authors:** Hoontaek Lee, Junsoo Kim, Kumjae Shin, Wonkyu Moon

**Affiliations:** 1Department of Mechanical Engineering, Phohang University of Science and Technology (POSTECH), Pohang-si 37673, Korea; lht1123@postech.ac.kr (H.L.); chak7258@postech.ac.kr (J.K.); 2Safety System R&D Group, Korea Institute of Industrial Technology (KITECH), Gyeongsan-si 38408, Korea

**Keywords:** scanning probe microscopy (SPM), scanning capacitive microscopy (SCM), FET sensor, subsurface imaging

## Abstract

We report recent improvements of the tip-on-gate of field-effect-transistor (ToGoFET) probe used for capacitive measurement. Probe structure, fabrication, and signal processing were modified. The inbuilt metal-oxide-semiconductor field-effect-transistor (MOSFET) was redesigned to ensure reliable probe operation. Fabrication was based on the standard complementary metal-oxide-semiconductor (CMOS) process, and trench formation and the channel definition were modified. Demodulation of the amplitude-modulated drain current was varied, enhancing the signal-to-noise ratio. The *I*-*V* characteristics of the inbuilt MOSFET reflect the design and fabrication modifications, and measurement of a buried electrode revealed improved ToGoFET imaging performance. The minimum measurable value was enhanced 20-fold.

## 1. Introduction

In the time since the atomic force microscope (AFM) was invented in 1986 [1], scanning probe microscopes (SPMs) have been used to study surface topographies from the micrometer to the atomic scale. SPM maps the distributions of physical properties by simultaneously measuring tip–sample interactions and the topography [2,3,4,5,6]. Various SPM techniques are used for electrical measurements, including electrostatic force microscopy (EFM) [2,7,8], Kelvin probe force microscopy (KPFM) [3,9,10], scanning capacitance microscopy (SCM) [4,11,12], and scanning spreading resistance microscopy (SSRM) [13,14,15]. These techniques measure contact potential differences [3,10], dopant concentration profiles [7,9], and local capacitance [8,12]. The SPM is affected by the cantilever excitation method used, signal processing, and the tip coating. Most SPM techniques used to derive electrical measurements require a conductive tip. Commercial SPM devices for the electrical measurement use probes of the same construction. Only the stiffness of the cantilever and the type of metal coated will change as appropriated for the application. To overcome the limitations that arise from the same construction, dedicated probes for electrical measurements employing electronic devices have been reported but have not yet been commercialized. In the 2000s, a resistive probe used the sample to serve as the gate, inducing the field effect at the tip [16]. The probe was used to measure surface potentials [17] and to define the ferroelectric domains of piezoelectric materials [18,19]. Additionally, a probe in which the substrate served as the gate was devised. With the aid of a nanowire field-effect transistor (NWFET), the surface potential of an electrode pattern was imaged in a non-contact manner [20].

We also developed a new probe based on a metal-oxide-semiconductor field-effect-transistor (MOSFET) for evaluation of various electrical properties. We term this the tip-on-gate of field-effect-transistor (ToGoFET) probe. We studied polarization of piezoelectric materials [21] and electrical damage caused by ion milling [22]. Recently, the local capacitances of dielectric films have been studied [23,24]. The ToGoFET probe is very sensitive because it incorporates a MOSFET. Additionally, the electrode that interacts with the sample is located at the end of the cantilever and is thus not affected by parasitic capacitance between the lever and the sample. However, the MOSFET gate was leaky [24], and the resulting variable substrate voltage compromised measurement performance and stability. In addition, the flicker noise inherent in a MOSFET [25] negatively affected performance. The flicker noise appears prominently in electronic devices, which is generated by modulation of surface potential by trapping and de-trapping of charges continuously occurring at the interface between SiO_2_ and Si and fluctuations in bulk mobility itself [26]. Thus, we improved the ToGoFET probe for the issues in terms of capacitive measurements. We modified probe design and fabrication, and the signal processing. We first describe the improved design and fabrication, and then the demodulation changes. We quantitatively evaluated the performance improvements using the *I*-*V* characteristics of the inbuilt MOSFET and the images obtained with the new probe. All comparisons are based on the most recent work on the old probe [24]. The new ToGoFET probe exhibits an improved signal-to-noise ratio and a reduced input voltage requirement, expanding the field of application.

## 2. Re-Design of the ToGoFET Probe

### 2.1. Operating Principle

The sensing mechanism of the ToGoFET probe is based on the field effect. Figure 1 shows the structure of the probe. An inbuilt MOSFET is fabricated at the end of the cantilever. A metallic tip was prepared on the MOSFET gate; ideally, the tip and gate are equipotential. The tip electrical potential induced by the sample surface is transferred to the gate, generating the field effect. The substrate conductance is modified by the field effect, and the amount of current flowing between the drain and source reflects the field-effect strength. The electrical potential induced at the tip can be derived from the current flow. This sensing mechanism is identical to the operating principle of a MOSFET. In addition, the induced voltage is affected by sample impedance. The impedance between the metallic tip and the reference electrode of the sample (which is patterned on the surface or might be an SPM plate) depends on both the material and its structure, and the induced voltage thus varies by the material and its structure because the impedance is connected in series with the gate oxide. Therefore, in the overall probe-sample system, the electrical properties of the current flow are determined by the material and its structure near the contact.

### 2.2. Structural Design Modification

The ToGoFET probe has been used to study polarization of a piezoelectric material [21], dielectric properties of oxide films [23,24], and ion damages on the semiconductor material [22]. Probe stability has gradually improved, but remains of concern. Sensitivity was unstable and the gate oxide sometimes exhibited dielectric breakdown. The inbuilt MOSFET was fabricated on a silicon-on-insulator wafer lacking a direct electrical contact with the substrate (i.e., the MOSFET body). Although an indirect electrical connection ran through the drain or the source, the substrate was not perfectly grounded. Therefore, the electrical potential fluctuated, changing the MOSFET operating point, thus rendering sensitivity unstable. Therefore, we added a body contact to ensure electrical stability. In addition, the new MOSFET features dimensional margins that avoid performance failures caused by dislocation or overlap. The values of critical design variables (the width-over-length ratio (*W*/*L*) and the gate oxide thickness) were not changed, and thus remain at 4 and 30 nm, respectively.

Another cause of instability was damage caused during fabrication of the conductive tip using a focused ion beam (FIB), whereby the beam damaged the gate oxide, causing leakage during capacitive measurements [24]. Therefore, the electrodes were separated by inter-layer dielectrics (ILD) and the contacts established through holes. Thus, the focus was on the stability of measurements and on the realization of accurate dimensions of the inbuilt transistor, which are essential for sophisticated evaluation of samples.

## 3. Fabrication

### 3.1. The Standard CMOS Process

The fabrication of MOSFET followed a standard complementary metal-oxide-semiconductor (CMOS) process while maintaining design parameters related to measurement performance. For example, the gate dimension and the implanted ion dose into the channel remained the same as those of the previous ToGoFET probe. The thickness of each layer of the 6-inch silicon-on-insulator (SOI) wafer was determined by reference to the required cantilever stiffness and the etching selectivity. A buried oxide layer (BOX) protects the device layer from over-etching, and BOX must withstand over-etching due to the uneven etch rate within the wafer during deep reactive ion etching (DRIE). Therefore, the device layer and the BOX were 5 and 2 μm-thick, respectively. Figure 2 shows the MOSFET fabrication process. Appropriate cleaning was performed between all steps.

To allow cantilever release, the ToGoFET probe was fabricated on an SOI wafer. First, the active area was defined via shallow trench isolation (STI) (Figure 2b). After filling the etched Si trench with SiO_2_, chemical-mechanical-polishing (CMP) was performed using an Si_3_N_4_ film as the stop layer. Phosphorus ion (P^+^) was implanted (at low energy, thus shallowly) into the active area to allow threshold voltage control (Figure 2c). The gate oxide (thickness 30 nm) was deposited via oxidation (Figure 2d) and poly-Si, then deposited on the gate oxide. The poly-Si gate was patterned via dry-etching (Figure 2e). The source and drain were defined using high-dose ion implantation (Figure 2f). The poly-Si gate was also heavily doped with phosphorus ions to block ion injection into the Si substrate. The overlap area was reduced by simultaneous ion doping on the source, drain, and poly-Si gate; in the previous work, gate was defined via a separate process from the source and drain [23]. High-dose BF^2+^ ion implantation was performed on another active area near the source, to ensure body contact (Figure 2g), and an ILD was then deposited for passivation of that area (Figure 2h). The ILD layer protected the gate from electrical damage during the subsequent micromachined electro-mechanical system (MEMS) process. After etching the contact window (Figure 2i), the CMOS process was completed via Al metallization (Figure 2j).

### 3.2. Cantilever Release

During ToGoFET imaging, the conductive tip remains in contact with the surface. Given this contact-mode operation, any lateral force distorts images. Therefore, the ToGoFET probe was designed to be a V-shaped cantilever affording good lateral stiffness. Figure 3 shows the fabrication of the cantilever. First, a wafer was thinned by polishing to a thickness of 500 μm, and this minimized the etched depth deviations during DRIE (Figure 3a). Placing the MOSFET at the end, ILD (SiO_2_) and device layer (Si) were etched into the V-shaped cantilever (Figure 3b). To protect the front side, a thick photoresist (PR) layer was coated after the cantilever shape was defined. After such PR passivation, an Al hard mask for DRIE was patterned onto the backside of the wafer. DRIE for Si deep etch proceeded in the absence of a PR strip (Figure 3c). The BOX was etched using the same mask composed of both Al and PR layers (Figure 3d). Finally, all PR layers were stripped after the Al layer for laser light reflection was deposited (Figure 3e–f).

### 3.3. Conductive Tip Fabrication

The SPM probe tip is generally created before the cantilever is released. If conductivity is required, the probe is metal-coated. However, ToGoFET probe fabrication imposes practical constraints on manufacturing. The fabricated tip size is several micrometers, and this limits the critical dimension (CD) of subsequent processing. In addition, the step sequence of CMOS fabrication cannot be changed. The tip must be fabricated after the CMOS process. Given the presence of a MOS, the thermal budget must be low to maintain dopant distribution. Plasma process-induced damage (PID) is also of concern [27]. Therefore, the ToGoFET probe tip must be carefully fabricated. Given the various constraints mentioned above, the tip was fabricated using an FIB technique. Although an FIB is generally used to mill micro-areas, it can be used for material deposition using an appropriate current in a precursor gas environment [28]. Figure 4 shows the tip fabricated via FIB-induced deposition (FIBID). The current during FIBID ranged from 48 to 1.5 pA.

## 4. Signal Processing and Measurement Set-Up

### 4.1. Amplitude Modulation

As described in Section 2.1, the ToGoFET probe measures the electrical impedance of the local region. In the capacitive measurement to obtain information about local capacitance, direct current (DC) operation is not appropriate due to the leakage current. The gate voltage of the inbuilt MOSFET is determined by a resistance of the sample, not a capacitance, because the impedance of capacitance is close to infinity at a DC condition. On the other hand, alternating current (AC) operation enables the probe to measure capacitive information of the sample surface. Figure 5 describes the AC operation of the ToGoFET probe. An amplitude of the AC gate voltage ($v_{g}$) is determined by the capacitances: gate capacitance, $C_{\mathrm{ox}}$, and tip-sample capacitance, $C_{s}$.
(1)$$v_{g}\left(x\right)=\frac{C_{s}}{C_{s}+C_{\mathrm{ox}}}\cdot v_{\mathrm{in}}=T\left(x\right)\cdot v_{\mathrm{in}}$$
where $T\left(x\right)$ is transfer function from the sample electrode to the gate at a location of x. Considering the impedance of the series connection, the larger the local capacitance (the thinner the dielectric film), the higher the gate voltage amplitude. The modulated signal is output in the form of drain current ($i_{\mathrm{ds}}$):(2)$$i_{\mathrm{ds}}\left(x\right)=G_{m}\left(V_{\mathrm{GS}}\right)\cdot v_{g}=T\left(x\right)G_{m}\left(V_{\mathrm{GS}}\right)\cdot v_{\mathrm{in}}$$
where $G_{m}\left(V_{\mathrm{GS}}\right)$ is the transconductance of the inbuilt MOSFET at a gate DC voltage of $V_{\mathrm{GS}}$. The drain current is converted to the voltage ($v_{\mathrm{am}}$) with a proper magnitude for the imaging. The *I*-*V* converter in Figure 6a is used for the conversion. The gain was set to 680 × 10^3^ V/A using a feedback resistance ($R_{f}$), and the DC operating point of the inbuilt MOSFET was controlled by a load resistance ($R_{L}$).
(3)$$v_{\mathrm{am}}\left(x\right)=T\left(x\right)R_{f}G_{m}\left(V_{\mathrm{GS}}\right)\cdot v_{\mathrm{in}}$$

### 4.2. Circuits for the Demodulation

To display two-dimensional images, demodulation was required to derive the amplitudes of the modulated AC voltages. Figure 6b,c shows the basic structure of the demodulation circuits for the amplitude-modulated signal ($v_{\mathrm{am}}$), the carrier signal ($v_{c}$), and the output signal ($V_{O}$) (the frequency of $v_{c}$ should be identical to that of $v_{\mathrm{in}}$). In previous studies [23,24], an envelope detection method (Figure 6b) was utilized to demodulate the ToGoFET signal, and electrical images were successfully obtained. However, the rectifying structure of the envelope detection scheme causes voltage drops at the diodes, and these compromise probe measurement sensitivity for the small signal. Additionally, the envelope detection scheme is based on the resistor-capacitor (*R*-*C*) response, where a long response time can distort images. The synchronous detection method lacks diodes and capacitors (Figure 6c). A single circuit multiplies the signals and synchronizes the carrier signal, and only the original signal is shifted (to its original frequency band) upon multiplication by the carrier signal. When sinusoidal voltage at a frequency of $\omega_{c}$ is input to the sample, the synchronous detection proceeds as follows:(4)$$\begin{array}{ll} V_{O} & =v_{\mathrm{am}}\cdot v_{c}=T\left(x\right)R_{f}G_{m}\left(V_{\mathrm{GS}}\right)\cdot V_{\mathrm{IN}}\sin\left(\omega_{c}t\right)\cdot V_{C}\sin\left(\omega_{c}t\right)\\ & =R_{f}G_{m}\left(V_{\mathrm{GS}}\right)V_{\mathrm{IN}}V_{C}\cdot \frac{1-\cos\left(2\omega_{c}t\right)}{2}\cdot T\left(x\right)\;\\ & \;\left(\mathrm{where}v_{\mathrm{in}}=V_{\mathrm{IN}}\sin\left(\omega_{c}t\right),v_{c}=V_{C}\sin\left(\omega_{c}t\right)\right) \end{array}$$

Low-frequency noise (1/f noise) is shifted to a frequency band near the operation frequency by the multiplication. Finally, a low-pass filter is used to obtain the desired signal (a term of $T\left(x\right)$). Therefore, the images become clearer, as shown below.

### 4.3. Measurement Set-Up

Figure 7 shows a schematic of the measurement set-up. We used SPA 400 and SPI 3800 instruments (Seiko, Chiba, Japan) for ToGoFET imaging. To allow capacitive measurements, an AC voltage was input to the sample electrode [23,24]. The amplitude and frequency thereof were controlled at appropriate levels with consideration of the sample electrical impedance. The AC voltage was transmitted to the gate of the ToGoFET probe, and the gate voltage induced a drain current. The electrical characteristics of the sample surface were amplitude-modulated and output as an amplified drain current, and the demodulator extracted electrical information from the amplitude-modulated signals and created SPM images.

## 5. Measurements

### 5.1. I-V Characteristics of Inbuilt MOSFET

To measure *I*-*V* characteristics, a test MOSFET was fabricated on the same wafer as the inbuilt MOSFET, and the structure was identical to that of the inbuilt MOSFET except that a gate electrode was used for external connection. Figure 8 shows the *I*-*V* characteristics of the MOSFET after the CMOS process concluded, and earlier results are also shown. Although the gate *W*/*L* ratios differ slightly (previous study 4.25; this work 4), the current level is greater than in the previous device. Since there was no DC bias in the input voltage in this study, the ToGoFET operated at a gate voltage of 0 V. Therefore, the sensitivity of the ToGoFET probe is proportional to the transconductance at a gate voltage of 0 V, and corresponding values are 412 and 177 A/V, respectively.

### 5.2. Sample Information

To emphasize the performance improvements, measurements were performed on the buried metal sample used in the last study [24] (Figure 9). Interdigitated Al electrodes (the sample reference electrodes) were patterned on an Si substrate. Then, a 400 nm-thick oxide layer was deposited via plasma-enhanced chemical vapor deposition (PECVD). The interdigitated electrodes are of width 10 μm and thickness 165 nm. The gap between the electrodes is 3 μm.

### 5.3. Frequency Response

For capacitive measurements, an AC voltage of appropriate frequency is required. Figure 10 shows the frequency response of the ToGoFET probe in contact with the dielectric layer over a biased electrode. If resistance affects the output during AC voltage operation, the *R*-*C* structure reflects the slope of the response [24]. Such a frequency dependence implies that the output signal is affected by leakage, which indicates that the capacitive measurement is not being performed properly. As shown in Figure 5, if the output signal is completely output in relation to capacitances ($C_{\mathrm{ox}}$ and $C_{s}$), the frequency response shows a gain independent of frequency. The flat response of the ToGoFET probe in Figure 10 proved that gate leakage was reduced, and the local capacitance dominated the ToGoFET signal.

### 5.4. Buried Electrode Imaging

The buried metal sample was scanned by applying a sample bias of frequency 100 kHz. Figure 11 shows the results utilizing the ToGoFET probe and envelope detection. Figure 11a shows the topographic image and Figure 11b–i show the ToGoFET images for different input voltages. The bright area contains the buried electrode where voltage is applied, and the dark area contains the buried electrode connected to the ground. As the input voltage decreases, the image is affected by noise. Especially, at an input voltage of 20 mV_pp_, noise dominates the image. The synchronous detection method enhances image quality, and noise is reduced. Figure 12 shows the results obtained using the ToGoFET probe and synchronous detection. Figure 12a shows the topographic image and Figure 12b–i show the ToGoFET images at different input voltages. A comparison with the previous images confirms that synchronous detection improves ToGoFET probe performance, especially the minimum measurable value. A quantitative comparison of the two sets of results is presented in the Discussion Section, along with a comparison with previous results.

### 5.5. Discussion

We analyzed the line profiles to explore the performance improvements afforded by the new ToGoFET probe. The output voltage of the ToGoFET was proportional to the input voltage, but noise existed regardless of the input voltage. Images obtained using low-input voltages were particularly affected by noise; indeed, the image in Figure 11b was dominated by noise. Therefore, the deviation from the clearest image ($v_{in}=$ 1 V_pp_) of the line profile served as a criterion of the signal-to-noise ratio, assuming that noise in the line profile of the clearest image could be neglected. The line profiles of the measured images at the same locations were normalized to their maximum contrasts (Figure 13a), and the deviations of line profiles were calculated by root-mean-square of the difference at each point constituting the line profiles. Figure 13b shows the deviations, which decreased as the input voltage increased. A deviation of the line profile of a previous study is also shown, and the profile deviation at an input voltage of 1 V_pp_ is shown in Figure 13b. The value lay at the deviation of the line profile obtained via envelope detection at an input voltage of 200 mV_pp_. Using synchronous detection, the value lay between the deviations of the line profiles at input voltages of 40 and 60 mV_pp_. Thus, to obtain a buried metal image of the same quality, the new probe reduced the sample bias to 20%; by changing the demodulator, this fell further to 5%. The probe signal-to-noise ratio therefore initially improved 5-fold, and then 20-fold upon adoption of synchronous detection. The design and fabrication modifications improved performance 5-fold and synchronous detection enhanced performance a further 4-fold. Table 1 summarizes the improvements in this study. The noise characteristics supported these results. A fast Fourier transform of the ToGoFET signal was performed using a TDS2024B device (Tektronix, OR, USA). Figure 14 shows the *I*-*V* converter output in the frequency domain measured prior to scanning the buried metal sample. The spectrum shows the need for amplitude modulation of the ToGoFET probe. In the frequency band, the 1/f noise (the so-called flicker noise) predominated. If the probe were to operate with a DC sample bias, the output signal would be obscured by flicker noise, and signal processing could not distinguish the ToGoFET signal from noise. However, as shown in Figure 14, a signal modulated at 100 kHz lay at a frequency remote from the low-frequency noise band, and noise-reduced signals were captured during demodulation. The demodulation methods were compared in terms of how effectively they reduced noise. During synchronous detection, noise reduction was more efficient because low-frequency noise was shifted to a higher frequency band by multiplying it by the carrier signal. Thus, low-noise images of high sensitivity were obtained.

## 6. Conclusions

We improved the ToGoFET probe in terms of capacitive measurement. In this study, two major improvement methods were applied. One was the performance improvement of the probe itself, and the other was the reduction of the noise. The former involved re-design and the use of a well-established process, and the latter involved a change in the demodulation method. The *I*-*V* characteristics of the inbuilt MOSFET and frequency response showed the improved performance of the probe itself. The transconductance of the inbuilt MOSFET increased from 177 to 412 A/V, and gate leakage was greatly reduced, eliminating the frequency dependence. The SPM imaging results confirmed the performance improvements by both methods. Measurements were obtained utilizing both envelope and synchronous detection, and quantitative comparisons performed in terms of line profile deviations caused by noise. Such deviations revealed the minimum measurable values, and the value improved 5-fold in the new probe. A further 4-fold improvement was achieved by changing the detection method to reduce the flicker noise of the inbuilt MOSFET.

These improvements in the minimum measurable values facilitate the assessment of smaller changes in sample surface properties and prevent undesirable changes in sample electrical states (because the applied voltages are now reduced). These findings expand the applications of the ToGoFET probe, and the results promise to be interesting.

## Figures and Tables

**Figure 1 micromachines-12-01303-f001:**
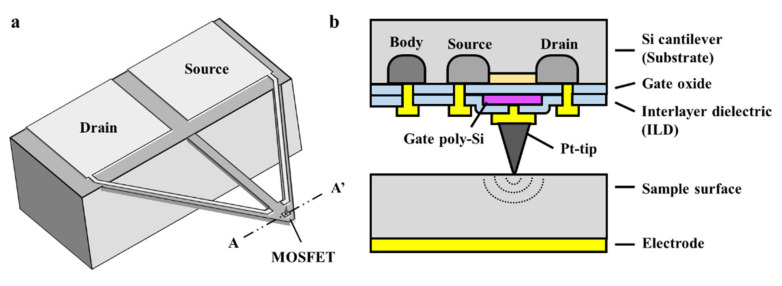
Schematic of the tip-on-gate of field-effect-transistor (ToGoFET) probe: (**a**) overall structure, and (**b**) cross-section of the sensing region. The inbuilt metal-oxide-semiconductor field-effect-transistor (MOSFET) is embedded at the end of the cantilever as a sensing component. The structure is identical to the typical planar MOSFET, and each electrode is connected to the pad along the V-shaped lever.

**Figure 2 micromachines-12-01303-f002:**
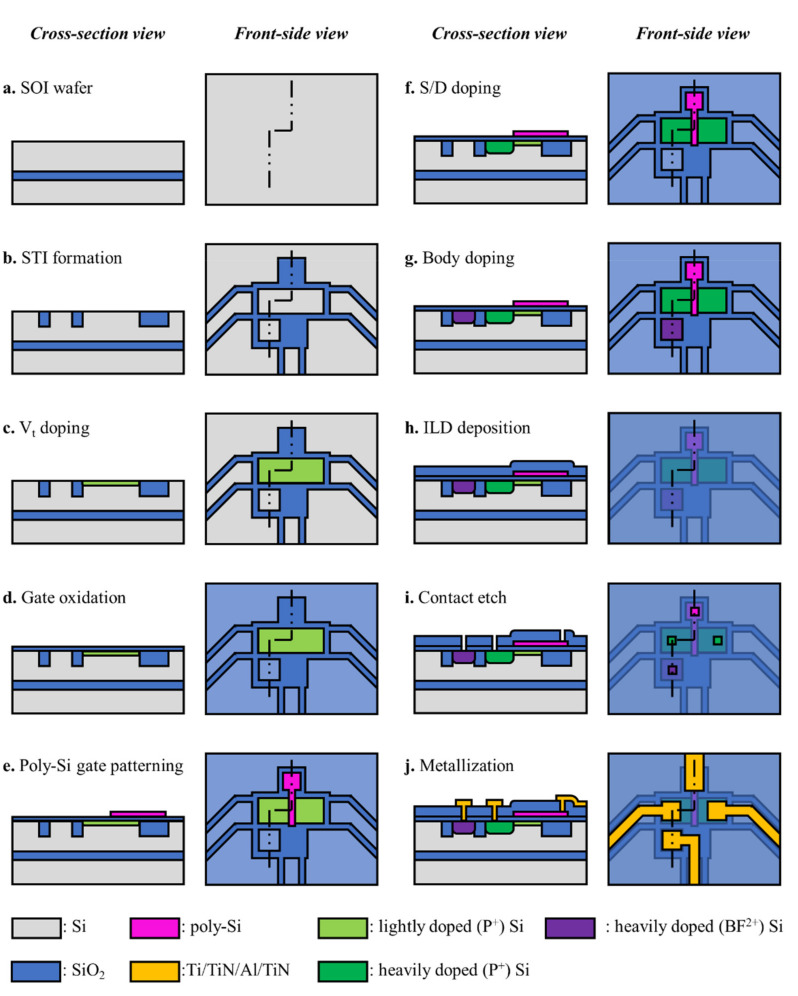
Fabrication of the inbuilt MOSFET: (**a**) The silicon-on-insulator (SOI) wafer was used for Si cantilever fabrication. (**b**) The active area of the device was defined by trench formation. (**c**) The shallow doping was performed to control the threshold voltage. (**d**) 30 nm-thick oxide is grown via oxidation. (**e**) Poly-Si was patterned as a gate. (**f**) Source and drain are heavily doped with P^+^ ions (the poly-Si gate was also doped in this process). (**g**) Body is heavily doped with BF^2+^. (**h**) Inter-layer dielectrics (ILD) was deposited on the entire surface. (**i**) The contact window through the ILD was etched for the electrical connection. (**j**) Metal electrodes (Ti/TiN/Al/TiN) connecting terminals to pads were patterned.

**Figure 3 micromachines-12-01303-f003:**
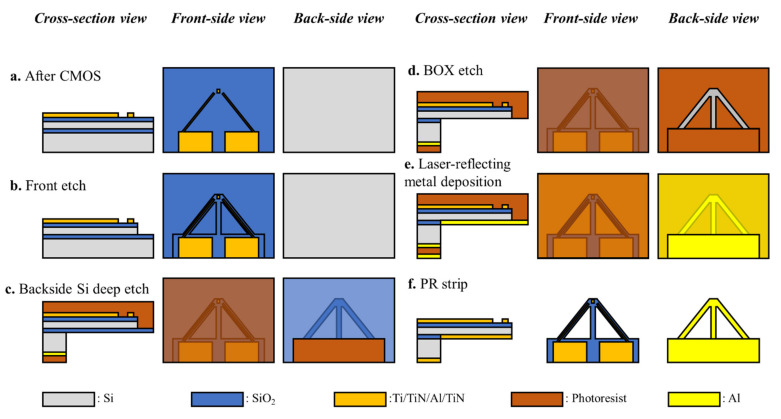
Fabrication of the V-shaped cantilever: (**a**) ILD and electrodes were exposed after the CMOS process. (**b**) Front-side was etched to the cantilever shape via the reactive ion etching (RIE) process. (**c**) Si deep etch via deep reactive ion etching (DRIE) process was performed from the backside of the wafer. (**d**) Buried oxide was etched via wet etching using the buffered oxide etchant (BOE). (**e**) Reflecting metal (Al) was deposited onto the backside of the cantilever. (**f**) Photoresists were stripped using acetone.

**Figure 4 micromachines-12-01303-f004:**
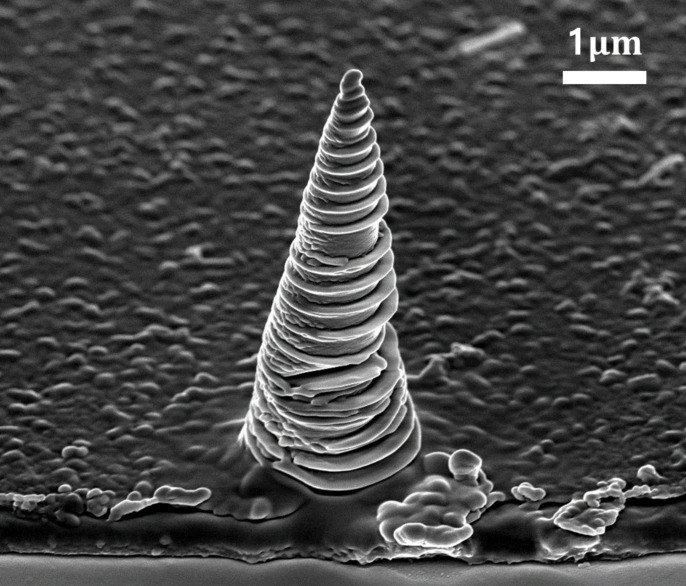
Electron micrograph of the Pt-tip produced via focused ion beam induced deposition (FIBID).

**Figure 5 micromachines-12-01303-f005:**
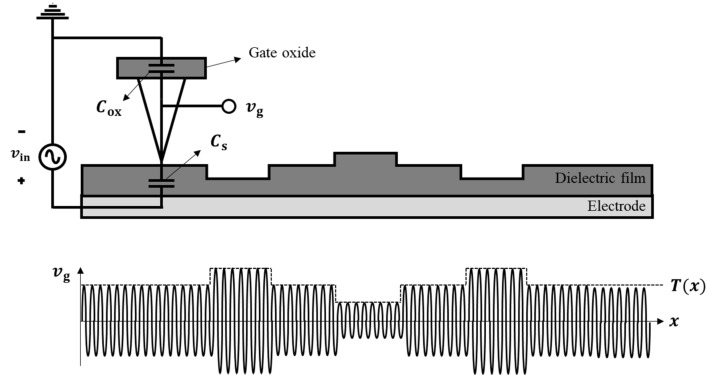
Concept of the amplitude modulation of the ToGoFET probe. The alternating current (AC) voltage to the sample modulates the electrical information of the sample surface.

**Figure 6 micromachines-12-01303-f006:**
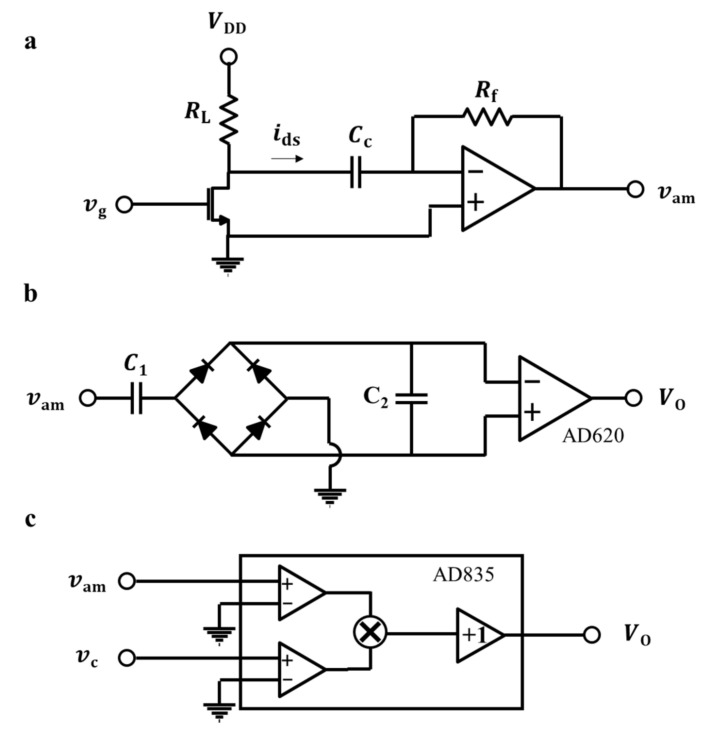
Basic structures of the circuits: (**a**) *I*-*V* converter, (**b**) envelop detection, and (**c**) synchronous detection.

**Figure 7 micromachines-12-01303-f007:**
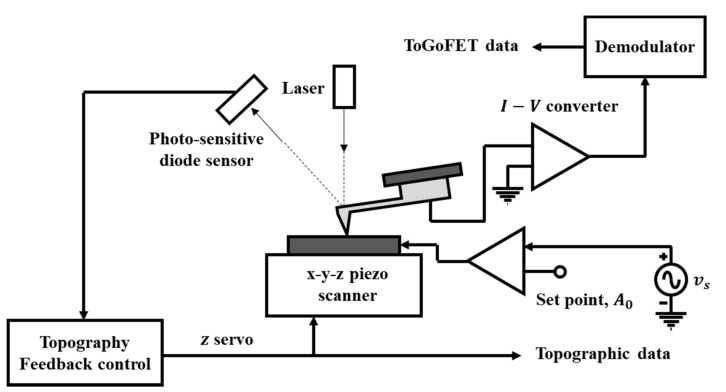
Schematic of the scanning probe microscopy (SPM) measurement set-up using the ToGoFET probe.

**Figure 8 micromachines-12-01303-f008:**
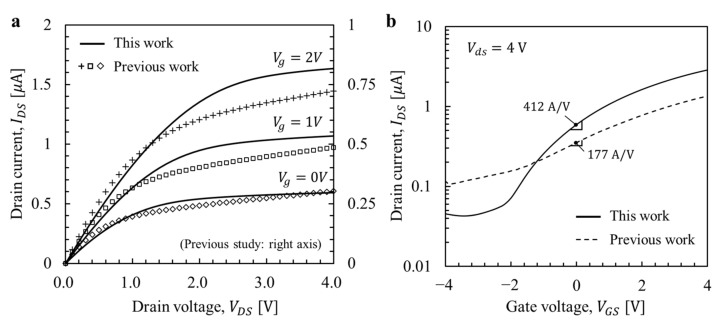
Comparison of *I*-*V* characteristics: (**a**) $I_{DS}$-$V_{DS}$ curves, and (**b**) $I_{DS}$-$V_{GS}$ curves.

**Figure 9 micromachines-12-01303-f009:**
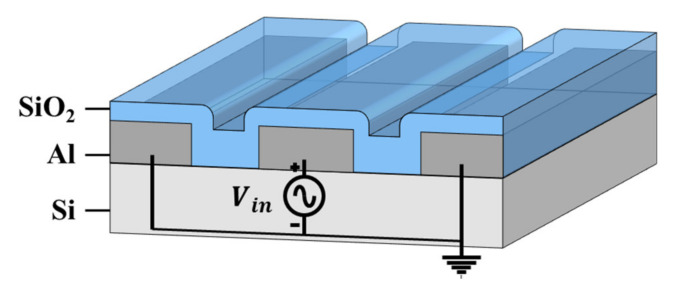
The buried metal sample used to analyze the performance of ToGoFET probes.

**Figure 10 micromachines-12-01303-f010:**
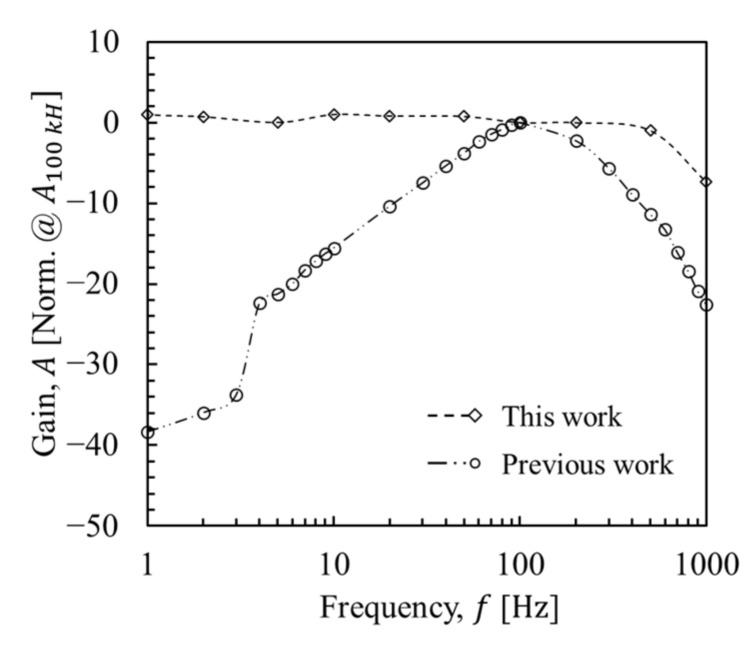
Frequency response of the ToGoFET probe in contact with an oxide film. The AC voltage was applied to the electrode pattern under the oxide film.

**Figure 11 micromachines-12-01303-f011:**
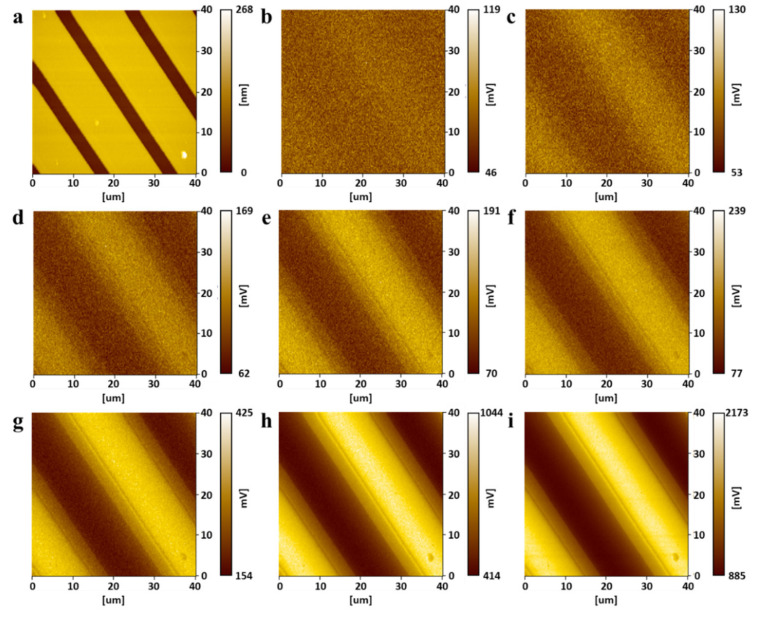
The ToGoFET images obtained using envelope detection: (**a**) topographic image, and (**b**–**i**) local electrical images for different input voltages ((**b**) 20, (**c**) 40 ${\mathrm{mV}}_{\mathrm{pp}}$, (**d**) 60 ${\mathrm{mV}}_{\mathrm{pp}}$, (**e**) 80 ${\mathrm{mV}}_{\mathrm{pp}}$, (**f**) 100 ${\mathrm{mV}}_{\mathrm{pp}}$, (**g**) 200 ${\mathrm{mV}}_{\mathrm{pp}}$, (**h**) 500 ${\mathrm{mV}}_{\mathrm{pp}}$, and (**i**) 1000 ${\mathrm{mV}}_{\mathrm{pp}}$).

**Figure 12 micromachines-12-01303-f012:**
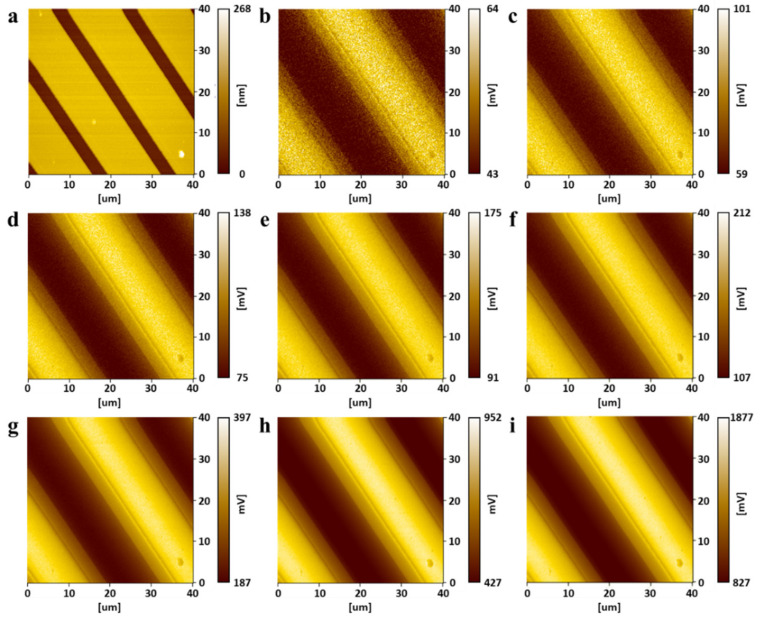
The ToGoFET images obtained using synchronous detection: (**a**) topographic image, and (**b**–**i**) local electrical images at different input voltages ((**b**) 20, (**c**) 40 ${\mathrm{mV}}_{\mathrm{pp}}$, (**d**) 60 ${\mathrm{mV}}_{\mathrm{pp}}$, (**e**) 80 ${\mathrm{mV}}_{\mathrm{pp}}$, (**f**) 100 ${\mathrm{mV}}_{\mathrm{pp}}$, (**g**) 200 ${\mathrm{mV}}_{\mathrm{pp}}$, (**h**) 500 ${\mathrm{mV}}_{\mathrm{pp}}$, and (**i**) 1000 ${\mathrm{mV}}_{\mathrm{pp}}$).

**Figure 13 micromachines-12-01303-f013:**
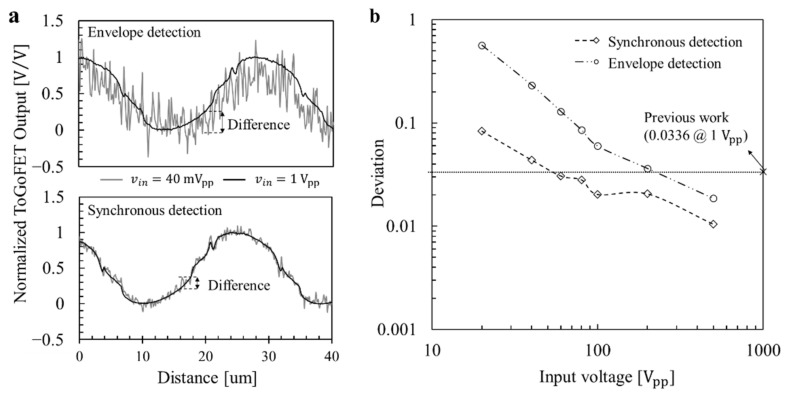
The deviations of the line profiles in ToGoFET images: (**a**) The differences when applying input voltages of 40 and 1000 ${\mathrm{mV}}_{\mathrm{pp}}$. (**b**) The deviations by the input voltage.

**Figure 14 micromachines-12-01303-f014:**
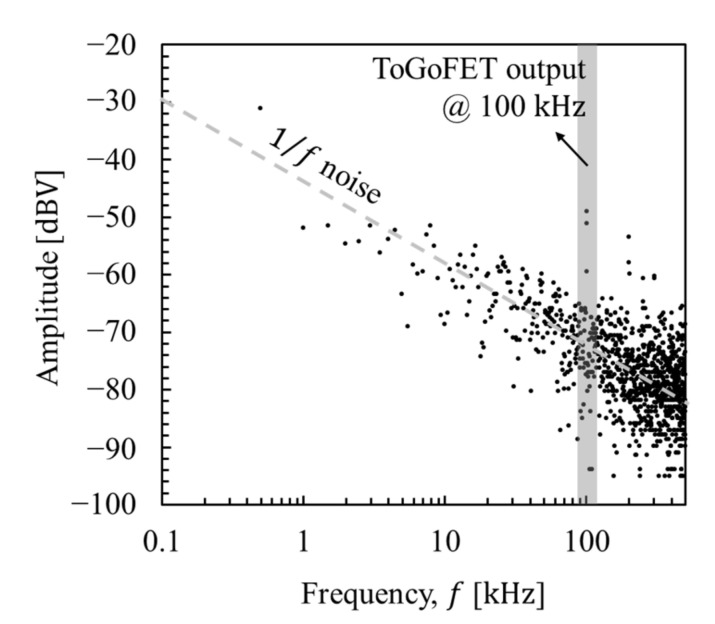
The ToGoFET signal in the frequency domain after the signal passed through the *I*-*V* converter.

**Table 1 micromachines-12-01303-t001:** Quantitative comparison with the previous research results.

	Methodology	Input Voltage for Same Line Profile Deviation	Note
Previous study	-	1000 mV_pp_ (Reference value)	Criterion for quantitative comparison.
1st approach	Modifications of Design and fabrication	200 mV_pp_	$G_{m}$(@ $V_{\mathrm{GS}}$= 0 V) was increased. (from 177 to 412 A/V) Gate leakage was decreased. (it was confirmed by frequency response)
2nd approach	Adoption of Synchronous detection	50 mV_pp_	Noise (deviations along the line profiles) was strongly decreased.

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
