# Peer review of "Improving the Performance of the ToGoFET Probe: Advances in Design, Fabrication, and Signal Processing"

_micromachines, 2021, doi:10.3390/mi12111303_

Round 1
Reviewer 1 Report
In this manuscript entitled ‘Improving the Performance of the ToGoFET Probe: Advances in Design, Fabrication, and Signal Processing’ by H. Lee and et al., a top gated field effect transistor (FET) is integrated onto an atomic force microscope cantilever in order to improve the probing in terms of capacitive measurement. In this study, the authors modified the design of the FET to reduce the noise in comparison with their previous published study at Sensors 2021, 21, (12), 4073; that results in an improvement experimentally. I recommend publication of this manuscript, a minor revision can improve the manuscript; please address the below comments.
- In the introduction part, please explain about ‘Flicker Noise’ and its origin
- In my view, the figure caption is very short and do not provide enough information for readers unless reader refer to the manuscript.
- In Figure 9, the gap between the electrodes is 3 um; did the author try with shorter gap such as 500 nm or less. If not, please explain why not.
Author Response
Thank you for your comments.
We tried to reflect your comment in the revised manuscript, and details on what are revised are found in the attached file.

Reviewer 2 Report
The manuscript by Lee et al., which proposes a method to improve performance of ToGoFET Probe in terms of design, fabrication, and signal processing, in comparison with a published one (also by this group), may contribute to the development of SPM and attract the readers in this field. Actually, the authors keep referring to another (their own) paper time to time in some sections for comparison. It is fine but does affect the coherence and cohesiveness of the manuscript (as a whole) as well as inconvenient for the readers to shuttle back and forth between both of them. The manuscript should be stand-alone coherent and cohesive. A quantitative table summarizing the indexes improved by the new method compared to the old ones can highlight its novelty/originality and make the manuscript stand alone on its own as well as easier for the readers to follow.
* Overall Recommendation: Accept after minor revision (corrections to minor methodological errors and text editing)
Author Response

(The authors gave the same response as above.)
